# Random Voids Generation and Effect of Thermal Shock Load on Mechanical Reliability of Light-Emitting Diode Flip Chip Solder Joints

**DOI:** 10.3390/ma13010094

**Published:** 2019-12-23

**Authors:** Jiajie Fan, Jie Wu, Changzhen Jiang, Hao Zhang, Mesfin Ibrahim, Liang Deng

**Affiliations:** 1Changzhou Xingyu Automotive Lighting System Co, Ltd., Changzhou 213022, China; wujie@xyl.cn (J.W.); dengliang@xyl.cn (L.D.); 2College of Mechanical and Electrical Engineering, Hohai University, Changzhou 213022, China; czjiang@sklssl.org; 3Changzhou Institute of Technology Research for Solid State Lighting, Changzhou 213161, China; 4EEMCS Faculty, Delft University of Technology, Delft 2628 CD, The Netherlands; hoayzhang@163.com (H.Z.); mesfin.ibrahim@connect.polyu.hk (M.I.); 5Department of Industrial and Systems Engineering, The Hong Kong Polytechnic University, Hung Hom, Hong Kong

**Keywords:** light-emitting diode, flip chip, solder joint, randomly distributed voids, reliability

## Abstract

To make the light-emitting diode (LED) more compact and effective, the flip chip solder joint is recommended in LED chip-scale packaging (CSP) with critical functions in mechanical support, heat dissipation, and electrical conductivity. However, the generation of voids always challenges the mechanical strength, thermal stability, and reliability of solder joints. This paper models the 3D random voids generation in the LED flip chip Sn96.5–Ag3.0–Cu0.5 (SAC305) solder joint, and investigates the effect of thermal shock load on its mechanical reliability with both simulations and experiments referring to the JEDEC thermal shock test standard (JESD22-A106B). The results reveal the following: (1) the void rate of the solder joint increases after thermal shock ageing, and its shear strength exponentially degrades; (2) the first principal stress of the solder joint is not obviously increased, however, if the through-hole voids emerged in the corner of solder joints, it will dramatically increase; (3) modelling of the fatigue failure of solder joint with randomly distributed voids utilizes the approximate model to estimate the lifetime, and the experimental results confirm that the absolute prediction error can be controlled around 2.84%.

## 1. Introduction

As it has advantages of high light efficiency [1], small volume [2], long lifetime [3], rapid response, and being environmentally friendly [4], light emitting diodes (LEDs) have become one of the most popular optoelectronic light sources and have been widely applied in the field of general lighting, displays, and communications, among others [5,6]. To fulfill the requirement of high efficacy, high color rendering, and small size, LED chip-scale packaging (CSP) with the flip chip solder joints has been proposed in recent years [7]. The LED CSP is composed of an GaN-based blue LED chip and a thin phosphor/silicone film adhered to the LED chip by the hot-pressing approach. Compared with traditional LED packaging, the CSP technology helps to shorten the packaging process time and accurately control the film thickness and color consistency [8]. Because the operation power, heat generation, and packaging density are becoming higher and the operation condition is severer, the demand on the highly reliable LED CSP is dramatically increasing [9,10]. For the package to be effective, the solder joint plays critical roles in electric conduction, heat conduction, and mechanical connection. Actually, because the LED CSP always inevitably suffers the thermal cycling and even shocks from outside environments, the mismatches of thermal expansion among all packaging components under these conditions always generate periodic stress and strain in the package level [11,12]. The increase of loads on a LED CSP has put forward more requirements on the reliability of a solder joint, because the fatigue failure of the solder joint under the thermal cycling condition will lead to the failure of whole package. Therefore, the thermal shock load on the mechanical reliability of the LED flip chip solder joints has always been one of the critical bottlenecks in the development of LED CSP technology [13,14,15].

Among all solder alloys, the lead-free Sn96.5–Ag3.0–Cu0.5 (SAC305) is considered as one of the most popular chip-attach candidates owing to its cost effectiveness, good solderability, and favorable mechanical strength [16,17]. It has widely been used in the flip chip soldering, but such solder connections are prone to failure over time under the thermal cycling [18] or mechanical bending test [19]. D. Kong et al. [20] used the Anand model to simulate the stress–strain response of the SnAgCu-based solder joint under the condition of thermal cycle loading and predicted its lifetime with the Manson–Coffin model. They found that the addition of a certain amount of Ce and Fe in the SnAgCu solder joint could significantly increase its fatigue lifetime. L. Zhang et al. [21] analyzed the stress–strain response of SnAgCu-nano Al solder joint in the flip chip ball grid array (FCBGA) device. Their results revealed that the maximum value of stress–strain was concentrated on the corner of solder joints, and the stress–strain value of the solder joint was obviously lower than that of the SnAgCu solder, which indicates that the nano Al could improve the reliability of the SnAgCu solder joint. In addition, as the existence of voids involved in the solder layer during the soldering process is inevitable, many studies have considered the void effects on the mechanical, thermal, and reliability problems within solder joints [22,23]. In detail, the stress concentration will be generated around these voids, which can lead to the decrease of electric conduction, heat conduction, and mechanical connection of solder joints [24,25]. C. S. Jiang et al. [26] investigated the size and position effects of voids within solder joints on the reliability of LED CSPs through both mechanical and thermal characteristics. They also studied the effect of the reflow soldering process on the bonding strength of a solder joint [14]. V. N. Le, et al. [18] used a 2D model to describe the random distribution of voids and simulated the dissipated plastic energy to access the damage. K. C. Otiaba, et al. [27] randomly generated the void morphology and predicted the maximum damage site of a solder joint under the thermal cycling test with finite element analysis (FEA) simulation. They found that voids can either influence the initiation or propagation of damage in the solder joints depending on the configuration, size, and location of voids. M. I. Okereke and Y. X. Ling [28] investigated the thermal resistance of solder interface materials by taking the 3D void morphologies into consideration. However, the current studies rarely consider the fatigue failure prediction modeling on solder joints with a 3D randomly distributed voids, which is much closer to the actual situation.

In this paper, the effects of 3D randomly distributed voids on the shear strength and heat dissipation of solder joints in LED CSPs aged under thermal shock test are firstly analyzed. The main stress distribution, thermal resistance, and fatigue lifetime of solder joints under thermal shock test are predicted by means of finite element (FE) simulations, and finally the actual experimental measurement results are used to verify the accuracy of the proposed prediction methods. The remainder of this paper is organized as follows. Section 2 introduces theories and methodologies used in this study. Section 3 presents the test samples and the thermal shock test designed for the test samples. Section 4 discusses the effect of randomly distributed voids on mechanical and thermal performances and fatigue failure of solder joints with both simulations and experiments. Finally, the concluding remarks are presented in Section 5.

## 2. Theory and Methodology

### 2.1. Anand Constitutive Model

The Anand model is mainly used to describe the constitutive relationship of materials at high temperatures. The model uses a single internal variable, which is related to dislocation density, solid solution strengthening, and grain size effect, to describe the macro impedance of plastic flow in the material. It can reflect the deformation behavior that relates to the strain rate and the temperature of viscoplastic materials, the historical effect of strain rate, strain hardening, dynamic recovery, and other characteristics [20,21].

The Anand model uses both the flow function (Equation (1)) and the evolution function (Equation (2)) to unify the creep and rate independent plastic behavior of solder joint, and its flow function is as follows:(1)$${\dot{\varepsilon }}_{p}=A\exp(-\frac{Q}{RT}){\left[\sinh(\xi \frac{\sigma }{s})\right]}^{1/m},$$
where $\dot{\varepsilon_{p}}$ is inelastic strain rate, *ξ* is stress multiplier, *A* is pre-exponential factor, *Q* is activation energy, *R* is universal gas constant (gas constant is 8.314J/(mol·K)), *T* is absolute temperature, *m* is strain rate sensitivity index, *s* is internal variable, *h* is strain hardening parameter, and *σ* is effective stress.

The evolution function is as follows:(2)$$\dot{s}=\left\{h_{0}{\left(\left|B\right|\right)}^{a}\frac{B}{\left|B\right|}\right\}{\dot{\varepsilon }}_{p},$$
where *h*_0_ is hardening/softening constant; *a* is strain rate sensitivity of hardening/softening; and *B* is *a* transient creep parameter, and its value is as follows:(3)$$B=1-\frac{s}{s^{*}},$$
where *s*^*^ is the saturation value of *s* at a given temperature and strain rate, which can be expressed as Equation (4):(4)$$s^{*}=\hat{s}{\left[\frac{{\dot{\varepsilon }}_{p}}{A}\exp(\frac{Q}{RT})\right]}^{n},$$
where $\hat{s}$ and *n* are the coefficient for saturation value of deformation resistance and strain rate sensitivity, respectively.

The deformation impedance is proportional to the equivalent stress:(5)$$\sigma \;=\;cs;\;c\;=\;(T,\;\dot{\varepsilon_{p}}),$$
where *c* is material parameter and can be expressed at a constant strain rate:(6)$$c=\frac{1}{\xi }\sinh^{-1}{\left(\frac{{\dot{\varepsilon }}_{p}}{A}\exp\left(\frac{Q}{RT}\right)\right)}^{m}.$$

At a constant temperature and constant strain rate, the material will appear steady state, and saturated stress will appear at this time. The saturated stress can be obtained according to Formulas (4)–(6) and $\sigma^{*}=cs^{*}$:(7)$$\sigma^{*}=\frac{\hat{s}}{\xi }{\left[\frac{{\dot{\varepsilon }}_{p}}{A}\exp(\frac{Q}{RT})\right]}^{n}\sinh^{-1}\left[{\left(\frac{{\dot{\varepsilon }}_{p}}{A}\exp\left(\frac{Q}{RT}\right)\right)}^{m}\right].$$

At the same time, it can be converted to the stress value as follows:(8)$$\sigma =\sigma^{*}-{\left\{{\left(\sigma^{*}-\sigma \right)}^{\left(1-a\right)}+\left(a-1\right)\left[\left(ch_{0}\right){\left(\sigma^{*}\right)}^{-a}\right]{\dot{\varepsilon }}_{p}\right\}}^{1/\left(1-a\right)}.$$

As shown, the Anand constitutive model contains nine parameters, which can be directly entered into the finite element software to characterize the viscoplasticity of the material and carry out stress and strain analysis. The Anand constitutive model parameters of the SAC305 solder joint used in this paper are shown in Table 1.

### 2.2. Fatigue Failure Prediction Model

In this paper, the fatigue failure of a LED flip chip solder joint is predicted using the model proposed by Engelmaier [11]. This model was developed on the basis of the Coffin–Manson equation, mainly considering the influence of temperature parameters and thermal cycling frequency. The formula is shown as follows:(9)$$N_{f}=\frac{1}{2}{\left(\frac{\Delta \gamma }{2\varepsilon_{f}^{,}}\right)}^{\frac{1}{c}}.$$

In the equation, *N**_f_* is the failure times of samples, △*γ* is shear strain range of the solder joint (△*γ =*
$\sqrt{3}$△*ε*, △*ε* is the plastic strain range of the solder joint), *ε**^’^_f_* is the fatigue ductility coefficient, and *c* is the fatigue ductility index.
(10)$$c=-0.442-0.0006T_{s}+0.017\ln(1+f),$$
where *T**_S_* is the average temperature of solder joints at each cycle. *f* is the cycles for thermal shock per day, 1 ≤ *f* ≤ 1000 cycles/day.

## 3. Experiments and Tests

### 3.1. Thermal Shock Test

Owing to the mismatch of thermal expansion coefficients (CTE) between the chip, solder joint, and substrate, the thermal stress will be generated under the thermal cycle loading condition, which results in different deformation behaviors of different materials. As known, the thermal stress in the solder joint changes with the thermal cycling, and the cyclic thermal stress is considered as the fundamental reason for the fatigue failure occurring in the solder joint.

In this study, the blue LED CSPs were selected as research samples in which the blue LED chip was soldered on an aluminum Printed Circuit Board (PCB) substrate through the flip chip technology. As shown in Figure 1a, the chip specification is 2040 type. In the Figure 1b, the electrode pad material of the test sample is gold, the substrate material of the LED chip is sapphire, the solder pad material on the aluminum PCB substrate is copper, and the surface treatment is nickel-plated gold leaching. Figure 1c,d presents the 2D void distribution and 3D morphology of LED CSP solder joints after reflow soldering processing.

Referring to the JEDEC thermal shock test standard (JESD22-A106B), the thermal shock test condition selected in this paper is shown in Table 2. The temperature range is −40~150 °C, the conversion time from high temperature to low temperature is 60 s, and the conversion speed is around 38 °C/s. The specific operation procedure of the designed thermal shock ageing test is described as follows:

Step I: X-ray imaging was firstly used to detect the void rate and void distribution of all thirty-four samples. Then, six of them, marked as a29-34, were randomly selected and their shear strength of solder joints was measured by the DAGE4000 bond tester. The shear velocity was set as 0.3 mm/s. The remaining twenty-eight samples, numbered as a1-a28, were aged in the thermal shock test chamber (Mode: ESPEC TSG-201S).

Step II: after 400 cycles of aging, the aged samples were checked as to whether they could be lightened or not. If the sample could not be lightened, it would be judged as failed. Otherwise, the crack propagation in solder joints of all test samples was also checked by the X-ray imaging. Then, the shear strengths of seven samples, numbered as a1-a7, were measured.

Step III: the remaining test samples were aged continually and the measurement procedure was the same as the above step II.

### 3.2. Thermal Shock Test Result Analysis

Figure 2a shows the void rate in solder joints of blue LED CSPs under the −40~150 °C thermal shock ageing test. As shown, the averaged void rates at 0, 400, 800, 1200, and 1600 cycles were recorded as 3.73%, 3.88%, 4.65%, 4.73%, and 5.64%, respectively. Both the mean and variance of void rates grow with increasing of aging cycles, which can be related to the crack propagation occurring during the thermal shock ageing test.

Figure 2b describes the degradation of averaged shear strength of solder joint with aging cycles. As shown, the averaged shear strength of solder joint decreases with the increase of thermal shock cycles. In this study, an exponential function was assumed to fit the curve of the solder joint’s averaged shear strengths versus ageing cycles. As the uncertainties in both voids measurement and shear force measurement are large, the relationship between the void ratio increase and shear force decrease is not clearly matched. According to some previous studies [29,30,31], the intermetallic compounds (IMCs) at the interfaces between the solder joint with electrode pad and substrate can determine the shear strength of the solder joint. A complicated competition between voids and IMC is assumed to occur during the thermal shock ageing test, in which the voids generation will limit the growth of the IMC layer [31], and the growth of the IMC layer can change the composition of the solder joint, resulting in cracks and more voids [29]. In addition, when the shear strength dropped to 15% of the initial value, the samples were assumed as failed in this study. According to the fitting results of measurement shear force data, the fatigue failure lifetime of the solder joint in this experiment, *t**_f_*, can be estimated to be 5347 cycles.

## 4. Results and Discussion

### 4.1. Three-Dimensional (3D) Modeling

In this section, the CATIA 3D modeling software was firstly used for the modeling of LED CSP, as described in Figure 1. In the simplified model, the solder pad and surface coating of LED chip were neglected. The simplified LED model includes the LED chip, SAC305 solder joint, and substrate, whose materials are considered as isotropic. The LED chip and substrate are considered as homogeneous bodies, and the model is a centrally symmetric structure. Figure 3a is the 3D model of the LED blue light chip with solder joints and Figure 3b is the X-ray image of the solder joints.

The geometric dimensions of components in the 3D model are listed in Table 3. Among them, the chip size is 1 × 0.5 × 0.14 mm. The origin of modeling and simulation is located in Figure 3a, the XY plane of coordinate system is coplanar with the bottom surface of solder joints, the XZ plane of coordinate system is coplanar with the front surface of LED chip, and the solder joint is symmetrical in the YZ plane of coordinate system.

According to the X-ray image of the solder joint shown in Figure 3b, the void diameters and the void center coordinates X and Y can be extracted. Table 4 shows the numbers of test sample measured at each test cycle and the averaged void numbers and void rates corresponding to each solder joint. Herein, the right solder joint in Figure 3b is selected to explain the statistical method used in accounting the location and the diameter of voids. In this study, the statistical functions were used to describe the distributions of void diameter and void center position coordinates. Specifically, the lognormal distribution and normal distribution were used to fit the void diameters and X-Y coordinates, respectively. Figure 4 shows the statistical curve fitting results of void diameters and locations after 800 test cycles.

On the basis of the above statistical information collected from the experiment, this section establishes the random voids in the 3D model of solder joints; the procedure is described as follows:

Step I: generate a random void diameter in the XY plane using the lognormal distribution extracted in Figure 4a; then, set the boundary condition R < X < 0.425–R; R < Y < 0.5–R to limit the range of X and Y; next, randomly generate the value of X and Y according to the fitted normal distribution curves shown in Figure 4b,c. Then, the first void is established.

Step II: the idea of generating the second void is the same as the first one, but the bound condition, in which the sum of radius of the two voids must be larger than the distance between the two centers, should be satisfied.

Step III: the third void is generated to ensure no interference with the above generated two voids until the total voids in the XY plane are generated.

Step IV: consider the void position in the Z axis. As the X-ray image of voids used in this study is a 2D image, the Z coordinates cannot be determined. Considering that the thickness of solder joint is thin, the void position in Z axis is assumed to be randomly distributed within solder joints.

Step V: Python program was finally used to achieve the above process. The constraint condition is that the average error of each void rate between modeling and X-ray measurements should be controlled within ±0.1.

### 4.2. Mechanical Strength Analysis

After completing the construction of the 3D model for LED CSP considering random voids, this section investigates the mechanical strength of solder joints based on the finite element simulation. The first principal stress distribution was simulated to characterize the mechanical strength of solder joints. In the simulation process, the fixed constraint was imposed on the bottom surface of the substrate, and a shear force with 20 N was applied to the side surface of the LED chip with the size of 1 × 0.14 mm. The finite element model was constructed using the SOLID185 element. Table 5 lists the material properties of components used in the 3D model.

According to the 3D modeling procedure, five sets of random voids were simulated in the 3D models at each test cycle. Figure 5 shows one of the first principal stress distributions of solder joints at 0 cycle, which indicates that the stress is always concentrated near the random void in solder joints. Table 6 summarizes the simulated first principal stresses in solder joints and their mean values at each thermal cycle. Figure 6 shows the simulated first principle stress of solder joints after the thermal shock test. As indicated, the first principal stress of the solder joint decreases sharply in the interval of 0~400 cycles and remains relatively constant after that.

Next, we analyzed the effect of void size and location on the first principal stress distribution of the solder layer. As shown in Figure 5, there is a large void located in the right solder joint; its coordinates of X and Y are 0.4623 mm and 0.0966 mm, respectively, and its radius is 0.0449 mm. As shown in Figure 7, when its location changes from X = 0.4623 mm to 0.36 mm, the simulated first principal stress decreases from 882.7 MPa to 264 MPa. This result indicates that, when the void diameter is larger than the solder joint’s thickness (0.05 mm), which means it is a through-hole in solder joints, the first principle stress of the solder joint can be obviously influenced by the void position.

### 4.3. Heat Dissipation Analysis

This section investigates the heat dissipation performance of a solder joint with randomly distributed voids using the finite element simulation. The 3D simulation model is the same as the above mechanical simulation model. The finite element model was constructed by SOLID90 element and the intelligent grid division method. The convection coefficient between the model and air is set as 10 W/(m^2^·°C), the ambient temperature is 25 °C, and the lower surface of the substrate is controlled at 30 °C. The electric power of the LED blue light chip is 0.6 W and the volume of the chip is 0.07 mm^3^. The thermal power of the chip is estimated as 0.5334 W and the heat power of the unit volume is calculated as 7.62 (K/mm^3^). As required for the simulation of heat dissipation, Table 7 lists the thermal conductivities of all components used in the 3D model.

Figure 8a displays the temperature distribution of solder joints in the one LED CSPs at the 0 cycle condition. It is shown that the highest temperature of the solder joint is always located in the middle upper surface of solder joints and the lowest temperature is located at the outer corner of their bottom surfaces. Figure 8b is a Z axis cross section of the solder joints shown in Figure 8a, where the Z coordinate value of this plane is 0.025 mm; it also shows the temperature decreases from inside to outside of the solder joints.

To investigate the void effect on the heat dissipation of solder joints, the temperature distributions of solder joints under each thermal shock test condition were simulated by taking the void rates into consideration, and then the thermal resistances of solder joints during thermal shock test were estimated according to Equation (11). Table 8 lists the temperature difference (*T*_max_ − *T*_min_) of each solder joint and their averaged values at different ageing times.


The thermal resistance can be expressed as follows:
(11)$$R_{th}=\left(T_{\max}-T_{\min}\right)/P,$$
where *T*_max_ and *T*_min_ is the lowest temperature and highest temperature of the solder joint, respectively, and *P* is the thermal power. On the basis of Equation (11), the averaged thermal resistances at different ageing times are obtained and shown in Table 9. The simulated thermal resistances are demonstrated in Figure 9, in which we can see that, as the void rates grow with the increasing of aging cycles, the mean and variance of thermal resistances of the solder joint show the same increasing trend as void rates.

### 4.4. Fatigue Lifetime Estimation

By using the Anand and Coffin–Manson models described in Section 2.2., the fatigue failure simulation and lifetime prediction for solder joints of LED CSPs under thermal shock test are studied in this part.

Firstly, the a22–a28 samples that undergo the 1600 cycles thermal shock test were selected as the research objects. The diameter of voids and the X and Y coordinates in samples were calculated based on the measurements at the 0 cycle condition. The void measurement results are list in Table 10. The statistical curve-fitting and random void generation were kept consistent with the above methods used in the mechanical and thermal performance analysis. The lognormal distribution curve-fitting of void diameters and the normal distribution curve-fittings of X and Y coordinates are shown in Figure 10.

Then, the finite element model of the solder joint was constructed by the VISCO107 element and the finite element models of PCB substrate and LED chip were constructed by the SOLID95 element. Table 11 shows the material parameters of the PCB substrate and LED chip. Considering that the thermal shock has little effect on the PCB substrate and LED chip, the material parameters used in the finite element simulation were assumed as constant. However, the temperature-dependent material parameters of the solder joint are shown in Table 12. The zero displacement constraint in the XYZ three directions was applied to the bottom surface of the PCB substrate. The thermal cycling load remained consistent with the ambient temperature. The reference temperature was set at 300 K.

Next, the fatigue failure simulation of solder joints without voids was firstly conducted with the Anand constitutive model. The simulation result is shown in Figure 11a, in which the maximum Von Mises plastic strain was generated at the bottom outer corner of the solder joint, which is close to the PCB substrate. The time history curve of the Von Mises plastic strain at the maximum strain node is shown in Figure 11b. The plastic strain range △ε of solder joints without voids can be estimated as 0.006128. When it was inserted into the Coffin–Manson model (Equation (9)), the fatigue failure cycle of solder joints without voids, *N**_f_*, was calculated as 6307 cycles. The fatigue ductility index of SAC305, *ε**^’^_f_*, used in the Coffin–Manson model, is 0.24 [11]. The average temperature under thermal shock test was estimated as follows:(12)$$T_{s}=\left(T_{\max}+T_{\min}\right)/2,$$
where *T*_max_ and *T*_min_ represent the maximum and minimum temperature, respectively. The frequency of the thermal cycle *f* = 65.45 (cycle/d), so *c* = −0.40366.

Moreover, the Von Mises plastic strain was also simulated for the solder joints with randomly distributed voids. The Von Mises plastic strain simulation result from one of the test samples is shown in Figure 12a. The strain range was extracted from the same node of the solder joint model without void, and its Von Mises plastic strain time history curve is plotted in Figure 12b. Herein, according to the statistical curve-fitting results shown in Figure 1, we modeled 20 solder joints with the same void rate, but randomly generated voids. Their fatigue failure cycles were estimated using the Coffin–Manson model and then fitted by a two-parameter Weibull distribution (Equation (13)), as shown in Figure 13. According to Equation (14), the mean fatigue failure cycle of solder joints with randomly distributed voids, Mean Time to Failure (MTTF), was estimated to be around 5499 cycles.
(13)$$f(t)=\frac{\beta }{\eta }{\left(\frac{t}{\eta }\right)}^{\beta -1}e^{-{\left(\frac{t}{\eta }\right)}^{\beta }},$$
(14)$$MTTF=\eta \cdot \Gamma \left(\frac{1}{\beta }+1\right),$$
(15)$$e\%=\left|\frac{\left(MTTF-t_{f}\right)}{t_{f}}\right|\times 100\%,$$
where shape parameter *β* = 7.225 and scale parameter *η* = 5872.52 cycles were estimated with the maximum likelihood (ML) estimation. *Г* is the gamma function.

In a short summary, firstly, compared with the experiment result shown in Figure 3, the fatigue lifetime prediction absolute error, *e* %, is calculated based on Equation (15), around 2.84%, which means a high accuracy of fatigue failure prediction by inserting the 3D randomly distributed void model into the Anand and Coffin–Manson models can be achieved in this study. Secondly, by comparing the fatigue life estimation results for two cases with and without voids, it is found that the void will decrease the lifetime of solder joints. Therefore, improving the soldering process of LED CSP technology to reduce voids in solder joint is necessary for guaranteeing its high reliability.

## 5. Conclusions

To investigate the reliability of the SAC305 solder joint used in a LED CSP, the 3D modeling of solder joints with randomly distributed voids was firstly established in this study with statistical methods. The mechanical strength and heat dissipation performance of solder joints considering the effect of random voids were then investigated with finite element simulations. Finally, the thermal shock experiment and simulation were designed to evaluate the fatigue failure of solder joints in LED CSP. The results indicate the following: (1) after a long-term thermal shock test, the void rate of solder joints increases and its shear strength degenerates with an exponential trend; (2) as stress is always concentrated near the voids, the first principle stress of a solder joint is obviously influenced by the void position, especially when the through-hole voids are formed at the corner of solder joints; (3) when the void rate grows during the thermal shock aging test, the thermal resistance of solder joint also increases at the same time; (4) the formation of voids in solder joints always affects their reliability. The proposed lifetime estimation method by inserting the 3D randomly distributed void model into the Anand and Coffin–Manson models can increase the accuracy of fatigue failure prediction for solder joints in LED CSP.

## Figures and Tables

**Figure 1 materials-13-00094-f001:**
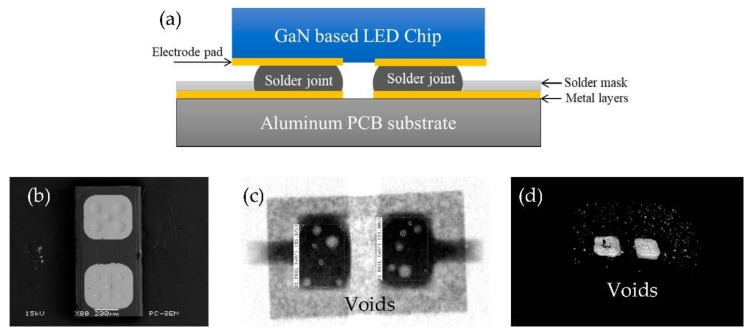
(**a**) The schematic diagram of test sample with a blue light emitting diode (LED) chip soldered on the aluminum PCB substrate; (**b**) the Scanning Electron Microscope (SEM) image of LED electrode pads; (**c**) the 2D and (**d**) 3D void distribution in the solder joints.

**Figure 2 materials-13-00094-f002:**
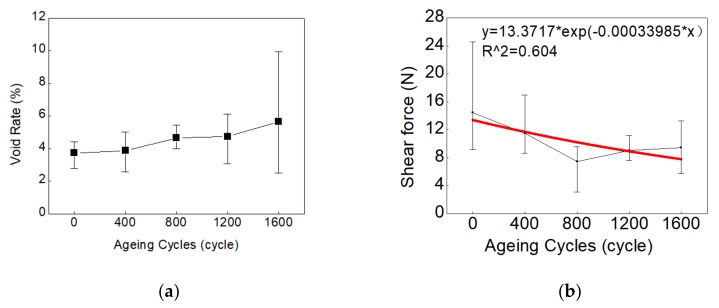
(**a**) The void rates in solder joints of LED chip-scale packaging (CSP) and (**b**) the shear forces of solder joints vs. ageing cycles.

**Figure 3 materials-13-00094-f003:**
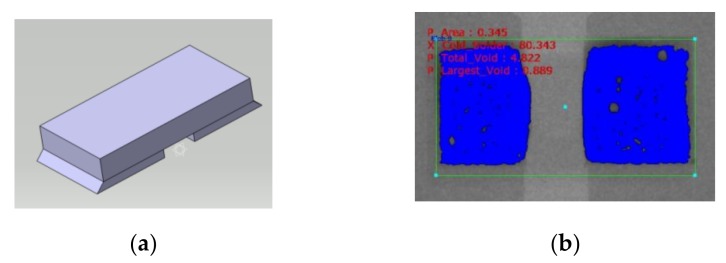
(**a**) Three-dimensional model with LED chip and solder joints and (**b**) X-ray image of the solder joints.

**Figure 4 materials-13-00094-f004:**
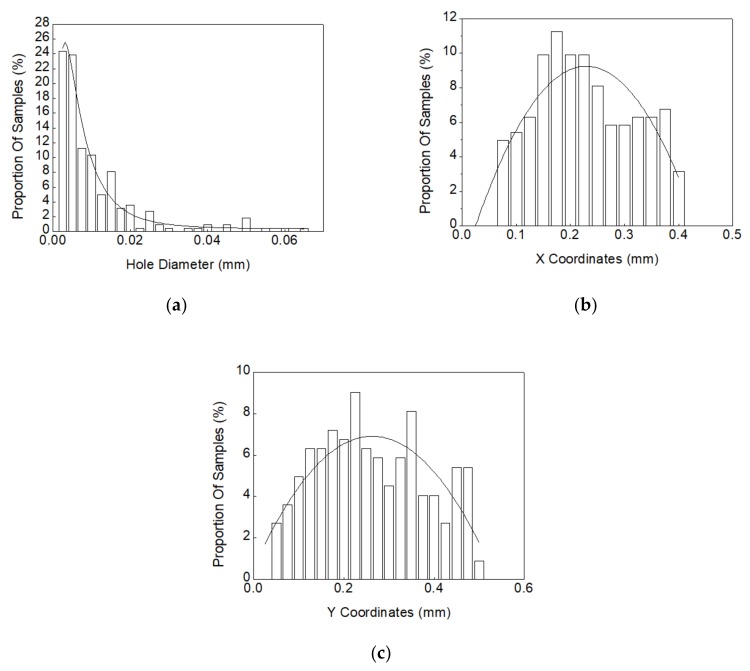
Statistical curve-fitting of void size and location distributions in solder joints: (**a**) lognormal distribution curve fitting of void diameters; (**b**) normal distribution curve fitting of X coordinates of voids; (**c**) normal distribution curve fitting of Y coordinates of voids.

**Figure 5 materials-13-00094-f005:**
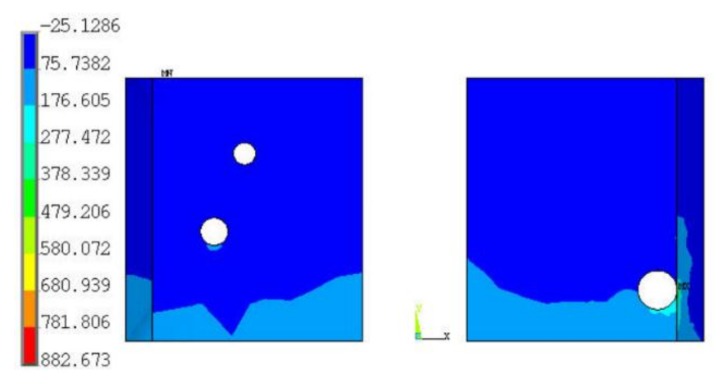
The first principal stress distribution of solder joints.

**Figure 6 materials-13-00094-f006:**
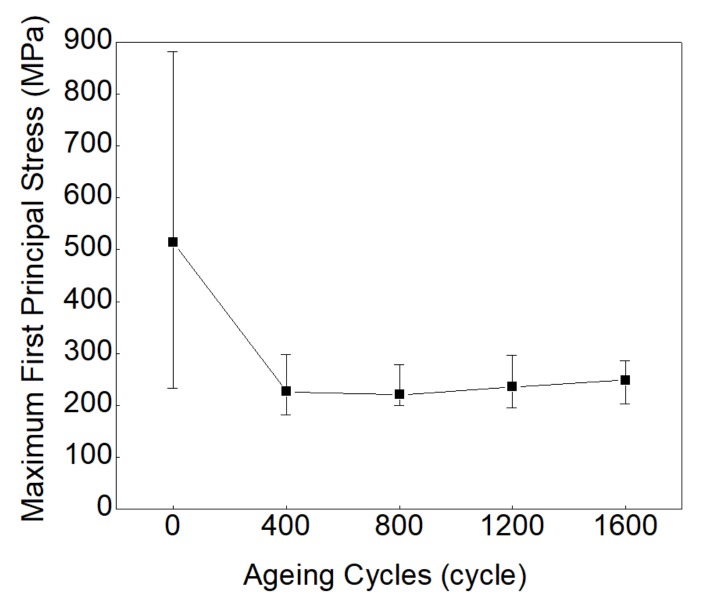
The simulated first principal stress of solder joints vs. ageing cycles.

**Figure 7 materials-13-00094-f007:**
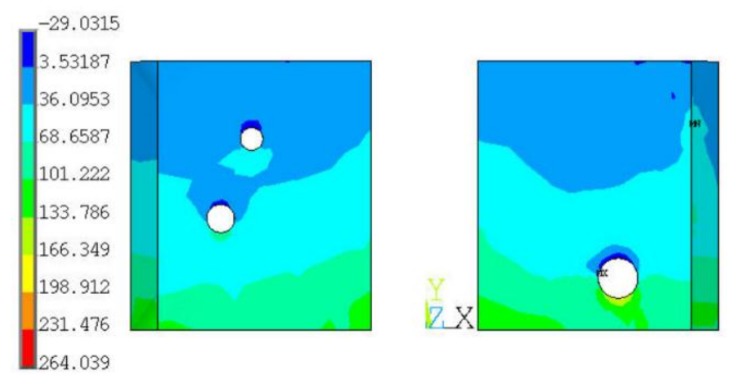
The effect of void size and location on the first principal stress distribution of the solder layer.

**Figure 8 materials-13-00094-f008:**
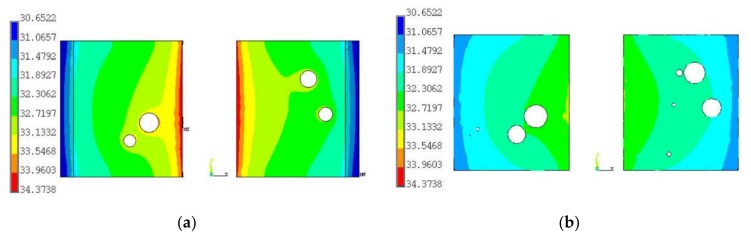
(**a**) The temperature distribution of solder joint; (**b**) the Z axis cross section of solder layers.

**Figure 9 materials-13-00094-f009:**
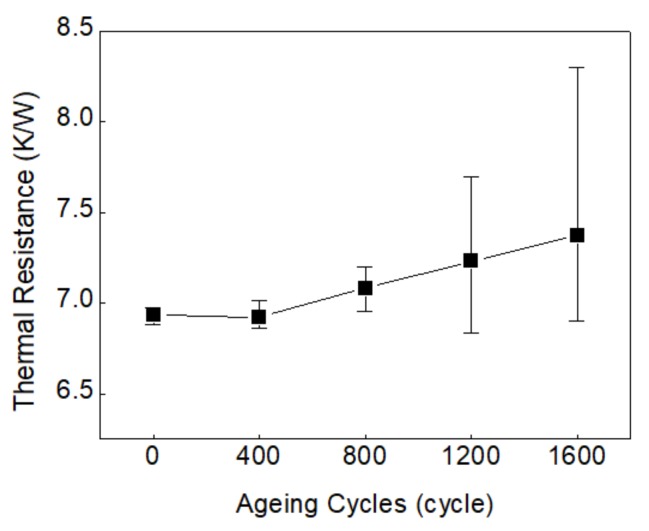
Thermal resistance of solder joints vs. ageing cycles.

**Figure 10 materials-13-00094-f010:**
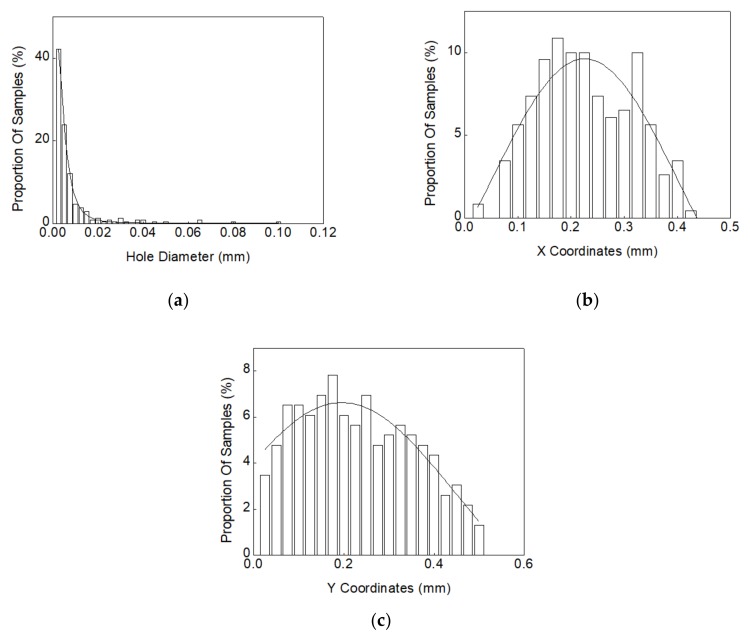
Statistical curve-fitting of void size and location distributions in solder joints of samples a22–a28: (**a**) lognormal distribution curve fitting of void diameters; (**b**) normal distribution curve fitting of X coordinates of voids; (**c**) normal distribution curve fitting of Y coordinates of voids.

**Figure 11 materials-13-00094-f011:**
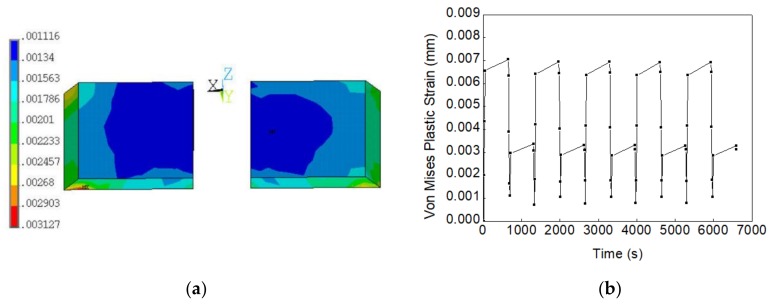
(**a**) Von Mises plastic strain distribution of void-free solder joints under thermal shock test; (**b**) Von Mises plastic strain of void-free solder joints vs. time.

**Figure 12 materials-13-00094-f012:**
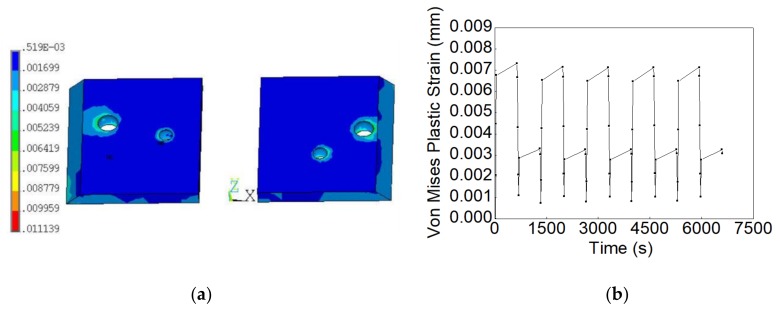
(**a**) Von Mises plastic strain distribution of solder joints with randomly distributed voids under the thermal shock test; (**b**) Von Mises plastic strain of solder joints with randomly distributed voids vs. time.

**Figure 13 materials-13-00094-f013:**
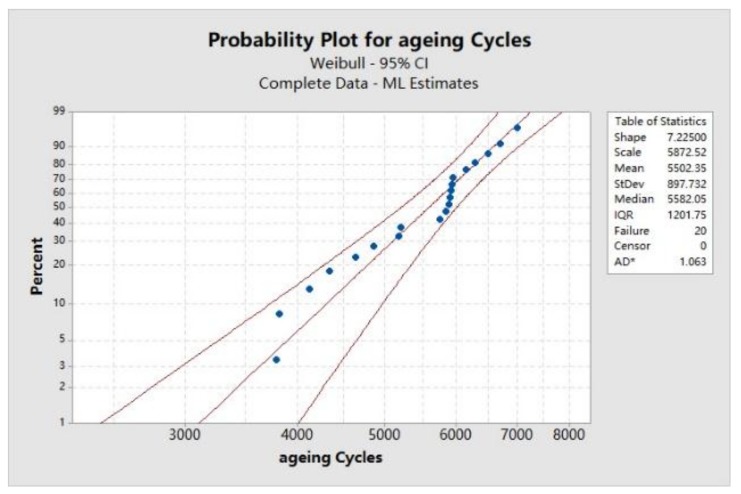
Weibull plotting of fatigue failure cycles. CI, confidence interval; ML, maximum likelihood; IQR, interquartile range.

**Table 1 materials-13-00094-t001:** The Anand model parameter setting for SAC305 solder [17].

Anand Constant	Units	Values	Descriptions
*S* _0_	MPa	45.9	Initial value of deformation resistance
*Q*/*R*	1/K	7460	Q = activation energy; R = universal gas constant
*A*	$s^{-1}$	5.87 × 10^6^	Pre-exponential factor
*ξ*	-	2	Stress multiplier
*m*	-	0.0942	Strain rate sensitivity index
*h* _0_	MPa	9350	Hardening/softening constant
$\hat{s}$	MPa	58.3	Coefficient for deformation resistance saturation value
*n*	-	0.015	Strain rate sensitivity of saturation (deformation resistance) value
*a*	-	1.5	Strain rate sensitivity of hardening/softening

**Table 2 materials-13-00094-t002:** Thermal shock test condition.

Temperature Range/°C	Peak Temperature Soaking Time/min	High and Low Temperature Conversion Time/s
−40–150	10	60

**Table 3 materials-13-00094-t003:** Geometric dimensions of the 3D model. LED, light emitting diode.

Materials	Area (mm^2^)	Thickness (mm)
LED chip	0.50	0.14
SAC305 solder joint	0.40 (up)/0.45 (bottom)	0.05
PCB board	400	0.94

**Table 4 materials-13-00094-t004:** The measured void information in the solder joint.

Thermal Shock Cycles	Numbers of Test Sample	Average Void Amounts	Void Rates %
0	6 (No. a29–a34)	36	3.92
400	7 (No. a1–a7)	32	3.84
800	7 (No. a8–a14)	30	5.20
1200	7 (No. a15–a21)	36	5.25
1600	7 (No. a22–a28)	40	5.81

**Table 5 materials-13-00094-t005:** Material properties of components used in the 3D model.

Components	Young’s Modulus/(Pa)	Poisson Ratio	Shear Modulus/(Pa)
LED chip	3.5 × 10^11.^	0.28	1.37 × 10^11.^
SAC305 solder joint	3.43 × 10^10^	0.32	1.29 × 10^10.^
PCB board	6.80 × 10^10.^	0.33	2.56 × 10^10.^

**Table 6 materials-13-00094-t006:** The first principal stress of solder joints.

No.	0 Cycle	400 Cycles	800 Cycles	1200 Cycles	1600 Cycles
M-Sim1#	280.5	185.8	220.2	194.5	274.6
M-Sim2#	233.3	203.3	278.6	215.3	202.3
M-Sim3#	331.1	297.3	203.1	296.8	213.4
M-Sim4#	840.9	181.2	198.8	245.9	265.0
M-Sim5#	882.7	264.2	200.1	223.3	285.4
Average	513.7	226.4	220.2	235.2	248.1

**Table 7 materials-13-00094-t007:** Thermal conductivities of components used in the 3D model.

Components	Thermal Conductivity (W/m·K)
LED chip	25
SAC305 solder joint	56
PCB board	194

**Table 8 materials-13-00094-t008:** The simulated temperature differences of solder joints.

No.	0 Cycle	400 Cycles	800 Cycles	1200 Cycles	1600 Cycles
T-Sim1#	3.71	3.66	3.71	3.76	4.43
T-Sim2#	3.72	3.73	3.81	3.91	3.68
T-Sim3#	3.72	3.67	3.84	4.11	3.73
T-Sim4#	3.68	3.74	3.73	3.86	3.72
T-Sim5#	3.67	3.67	3.79	3.64	4.10
Average	3.70	3.69	3.78	3.86	3.93

**Table 9 materials-13-00094-t009:** The averaged thermal resistances of solder joints vs. ageing cycles.

Thermal Shock Test (cycles)	0	400	800	1200	1600
Thermal Resistance *R*_th_/(K/W)	6.94	6.92	7.08	7.23	7.37

**Table 10 materials-13-00094-t010:** The measured void information in solder joints (samples a22–a28).

Numbers of Test Samples	Average Void Amounts	Void Rates %
7	32	3.733

**Table 11 materials-13-00094-t011:** Material parameters of the LED chip and PCB substrate.

Components	Modulus of Elasticity *E*/MPa	Poisson Ratio *μ*	Coefficient of Linear Expansion *a*_1_ (10^−6^ K^−1^)
LED chip	310	0.23	8
PCB substrate	68	0.33	23.4

**Table 12 materials-13-00094-t012:** Material parameters of the SAC305 solder joint.

Temperature/(K)	Modulus of Elasticity *E*/(MPa)	Poisson Ratio *μ*	Coefficient of Linear Expansion *a*_1_ (10^−6^ K^−1^)
233	45.71	0.32	19.1
271	39.03	0.32	19.1
309	32.35	0.32	19.1
347	25.67	0.32	19.1
385	18.99	0.32	19.1
423	12.32	0.32	19.1

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
