# Peer review of "Random Voids Generation and Effect of Thermal Shock Load on Mechanical Reliability of Light-Emitting Diode Flip Chip Solder Joints"

_materials, 2019, doi:10.3390/ma13010094_

Round 1
Reviewer 1 Report
This manuscript focused on Fatigue Failure Prediction for LED Flip Chip Solder Joints. The results both simulation and experimental analysis shown that with the increase of thermal shock ageing cycles, the void rate of solder joint increase and its shear strength exponentially degrades. This work is very important as reported work has applications in semiconductor technology. For current and next generation, this kind of work is very important. This report is recommended to publish in materials after the following minor revisions.

Author Response
Dear Editor and Reviewers,
We greatly appreciate the comments and detailed reviews for our manuscript. These comments are valuable and helpful for us to improve the paper. We have carefully revised the manuscript, and also included a detailed list of revisions along with our response to the reviewer comments as follows. All changes have been marked in yellow in the revised manuscript.
Sincerely,
Corresponding author:
Jiajie Fan

Reviewer 2 Report
Reviewer’s Comment
The manuscript presents critical research area needed to advance the technology of CSP. However, the reviewer has the following concerns:
General:
The title and the three results presented should be more in agreement. The results should be confirming the title.
For instance, the result (1) is on “effect of thermal shock load on void formation and shear strength. The result (2) is on “effect of void nature (through or blind) on solder joint principle stress. The result (3) appears to be reporting on the accuracy of generating the randomly distributed voids.
With reference to the results which should be revised, the title of the manuscript should be “Random voids generation and effect of thermal shock load on mechanical reliability of flip chip solder joints”.
Otherwise, results underpinning the aim and objectives of the research (article title) need to be presented. It points out that the manuscript needs to be more focus on the subject it hopes to present.
Some sentences can be review for correctness.
Abstract:
Need to specify the thermal shock test employed. Can in-situ thermal load be used because the reviewer does not think that the standard thermal load used is developed for fatigue life prediction purposes? Is it not supposed to be used for qualification?
Is the statement “… (1) with the increase of thermal shock ageing cycle,…” not ambiguous? Is the cycle time reduced and produced more cycle per time or higher temperature is used or ramp rate is increased …?
What is meant by …”maximum first principal stress…”? is it magnitude of first principal stress? Reviewer thinks that first principal stress is the maximum and third is the minimum?
Authors need to provide analytical cause of the dramatic increase in principal stress with emergence of through-hole voids in the corner of solder joints.
The assertion in result (3) is not proved. Otherwise, the result (3) is not properly presented and thus the sense it intends to convey is lost. This sentence could be rephrased to reflect “Modelling of the fatigue failure of solder joint with randomly distributed voids utilised approximate models to the real-life models and the results obtained from experiment is …”.
The authors need to be more specific in the presentation of their results. They should state the prediction results from numerical and from experimental and quantify any variance between the two.
Introduction:
Numerous sentences do not read well and their meanings are lost. Some arguments are not properly presented – maybe language related. Structure of the introduction can be improved.
Experiment.Fig 1 is poor and difficult to understand. Legibility can be improved. The 2D and 3D void model can be shown better and clearer.
3.1 Thermal shock test:
Is the appropriate terminology used? Ramp rate, dwell time, etc.
3.2 Thermal shock test result analysis
The reviewer has concern with the results presented in Fig 2(a) and (b). The Fig (a) shows the void rate as a monotonic increasing function of ageing cycles but the 800 cycle is a critical point which suggests occurrence of the effect or interaction of other parameter(s) not considered in the voiding process. The figure shows no significant difference between the 800 and 1200 cycles. In fig (b) the interpolation of an exponential function to the data plot does not seem adequate. This is demonstrated by the poor R2 value of 0.604. The data plot may have been better modelled with a polynomial function as it decreased to a minimum at 800 cycles and increased afterwards. More cycles up to 2400 is needed to confirm the data trend. The reviewer does not see from the plot of Fig(b) how the shear force decreases with ageing cycles as stated by the authors.
More importantly, there appear to be a major parameter which influences the shear force and to some degree the voiding rate which the authors seem to have neglected.
The authors stated that the solder pad material on the aluminium PCB substrate is copper and the SAC305 solder is used. In soldering an intermetallic compound (IMC) is form at the joint interface between a tin-based solder and copper pad on substrate during soldering. The IMC compound increases in thickness during ageing. The thickness impacts the joint strength and reliability. The formation straightens the bonding but makes it more brittle…. . Critical thickness of IMC is yet to be confirmed.
So, to understand the relationship between the shear force and the ageing cycle, the IMC thickness should be considered. In addition, the relationship with the voiding rate should be considered too. More so as this is a materials journal.
The assumption that “when the shear strength dropped to 15% of the initial value, the sample were assumed as failed in this study” appears unscientific. Any reference or citation for this? Any scientific basis? How the authors estimated 5347 cycles for the fatigue failure lifetime is not clear?
Results and Discussion.4.1 3D modelling
The model used is too simplified to product realistic results. The solder (copper) pad cannot be neglected because there are significant different among the CTEs of copper pad, solder and aluminium substrate which impact on the fatigue response of the assembled CSP. The reviewer does not think that the LED chip and substrate should be considered as a homogeneous body because they are not. Such consideration will lead to erroneous simulation results.
Author Response

(The authors gave the same response as above.)

Round 2
Reviewer 2 Report
No comments.